# ExaGPT: Example-Based Machine-Generated Text Detection for Human Interpretability

## Abstract

Detecting texts generated by Large Language Models (LLMs) could cause grave mistakes due to incorrect decisions, such as undermining student's academic dignity. LLM text detection thus needs to ensure the interpretability of the decision, which can help users judge how reliably correct its prediction is. When humans verify whether a text is human-written or LLM-generated, they intuitively investigate with which of them it shares more similar spans. However, existing interpretable detectors are not aligned with the human decision-making process and fail to offer evidence that users easily understand. To bridge this gap, we introduce **ExaGPT**, an interpretable detection approach grounded in the human decision-making process for verifying the origin of a text. ExaGPT identifies a text by checking whether it shares more similar spans with human-written vs. with LLM-generated texts from a datastore. This approach can provide similar span examples that contribute to the decision for each span in the text as evidence. Our human evaluation demonstrates that providing similar span examples contributes more effectively to judging the correctness of the decision than existing interpretable methods. Moreover, extensive experiments in four domains and three generators show that ExaGPT massively outperforms prior interpretable detectors by up to +37.0 points of accuracy at a false positive rate of 1%. We will release our code after acceptance.

## 1 Introduction

LLMs can yield human-like texts in response to various textual instructions (OpenAI, 2023a; Touvron et al., 2023). Ironically, the powerful generative capability has resulted in various misuses of LLMs, such as cheating in student homework assignments and mass-producing fake news (Tang et al., 2023; Wu et al., 2023). Such abuse of LLMs has sparked the demand for discerning LLM-generated texts from human-written ones. Recent studies have developed LLM-generated text detectors with promising performance (Mitchell et al., 2023; Su et al., 2023a; Koike et al., 2024; Hans et al., 2024; Verma et al., 2024).

While LLM text detection can help prevent potential misuse of LLMs, misclassifications could lead to severe consequences. For instance, web content writers have recently been at risk of losing their careers because of false-positive classification (Gizmodo, 2024). In school education, incorrect detection results might ruin students' academic dignity (OpenAI, 2023b; Bloomberg, 2024). At the same time, it is extremely difficult, if not impossible, to develop a perfect detector with 100% accuracy in such real-world scenarios, and there remain edge cases where human-written texts can be misidentified as LLM-generated and vice versa. Thus, it is crucial to create a detector that provides interpretable evidence, allowing users to judge how reliably correct the detection results are (Tang et al., 2023; Ji et al., 2024).

Most detectors lack the interpretability of their decisions, outputting only binary labels of who authored the text. There are few studies on the interpretability of the detection. Gehrmann et al. (2019) color-highlighted the tokens with high probability under the predicted distribution of LMs. Mitrović et al. (2023); Wang et al. (2024) showed which part of a text contributed to a decision based on prediction shifts via perturbations to the text. Yang et al. (2023) provided the $n$-gram overlaps between the original text and re-prompted ones generated by LLMs. Here, humans intuitively judge whether a text is human-written or LLM-generated by assessing with which source it shares more *similar spans*, including verbatim overlaps and semantically similar spans (Maurer et al.,

2006; Barrón-Cedeño et al., 2013). However, current detectors are not aligned with the human decision-making process (Figure 1) and fail to yield sufficiently interpretable evidence for users.

Motivated by this gap, we present **ExaGPT**, an interpretable detection method based on the human decision-making process of verifying the origin of a text. In particular, ExaGPT makes a prediction by examining whether the text shares more similar spans with human-written vs. with LLM-generated texts from a datastore. This approach can provide similar span examples that contribute to the decision for each span in the text as interpretable evidence. To present interpretable span-segmented text as a final result, we apply a dynamic programming algorithm and determine the optimal span break. It balances the long span length and its high frequency with the datastore (i.e., many similar phrases to the span exist in the datastore). The similarity of the retrieved spans to each span in the target text can help users judge the reliability of the detection result.

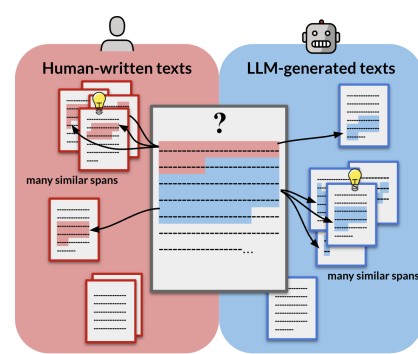

Figure 1: Identifying the author of a text (human vs. LLM) by examining if it shares more similar spans, including verbatim overlaps and semantically similar spans, with human-written vs. LLM-generated texts.

To evaluate the interpretability of LLM detection, we conducted a human evaluation of how well people can infer the correctness of the detection from the detector's evidence, and we found that providing similar span examples contributes more effectively to judging the correctness of the detection than existing interpretable methods. Moreover, extensive experiments in four domains and three generators showed that ExaGPT massively outperforms prior interpretable and powerful detectors by up to +40.9 points accuracy, even at a constant false positive rate of 1%. From these results, we observe that ExaGPT achieves high interpretability in its detection result and also high detection performance.

## 2 METHODOLOGY

ExaGPT classifies a text based on whether it shares more similar spans with human-written or with LLM-generated texts from a datastore. As a final result, ExaGPT offers the span-segmented text where each span is accompanied by similar span examples that contribute to the decision. Figure 2 illustrates the workflow of ExaGPT, which has two phases: **Span Scoring** and **Span Selection**. In the first phase, we mainly investigate whether each span in the target text shares more similar spans with human-written or LLM-generated texts from a datastore. Meanwhile, we calculate scores for each span, which we use in the second phase (§2.1). In the second phase, we primarily decide the optimal span segmentation to aid users' understanding of the final result. Specifically, we apply a dynamic programming (DP) algorithm with the scores from the first phase to find the span boundaries, balancing span length and its frequency within the datastore (§2.2). Finally, we detect the target text based on the selected spans and we provide similar span examples for each target span as evidence (§2.3). We will go into further details below.

### 2.1 SPAN SCORING WITH $k$-NN SEARCH

Given a target text $x$ to be classified, we define an $n$-gram span in the text $x$ as $x_{i:i+n}$, which is any continuous sequence of $n$ tokens starting in the $i$-th token. For each $n$-gram target span $x_{i:i+n}$, we retrieve the top-$k$ most similar[1] $n$-gram spans $s_j$ $(j \in \{1, \ldots, k\})$ from the datastore, with each original label and similarity $\{(s_j, l_j, c_j)\}_{j=1}^k$. Here, $l_j$ is Human when the span $s_j$ is part of a human-written text, or LLM when the span $s_j$ is a part of a LLM-generated text. $c_j$ is the similarity between the target span $x_{i:i+n}$ and each retrieved span $s_j$.

---

[1]We encode the target span, and all spans in the datastore into the same embedding space. We then perform $k$-nearest neighbor ($k$-NN) search based on the cosine similarity of each two span embeddings. See more details in §3.1.

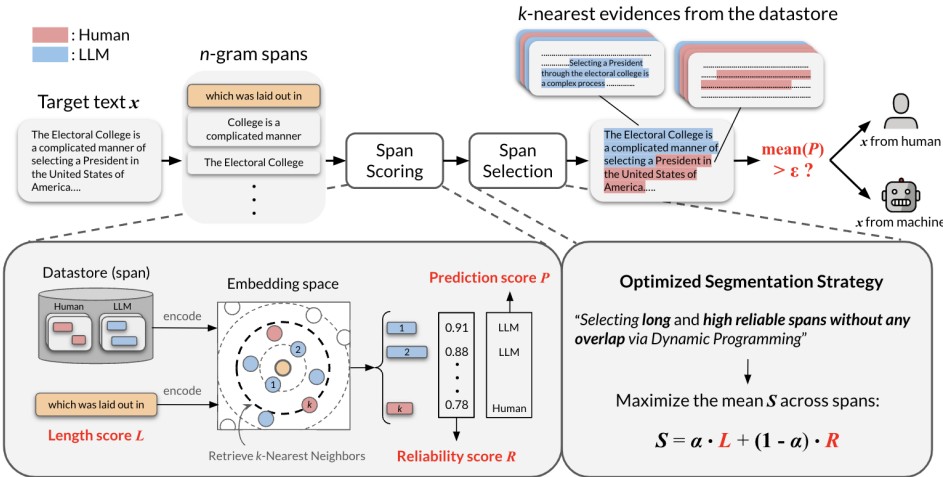

Figure 2: Overview of ExaGPT. It detects the author of a text by examining whether the text shares more similar spans with human-written texts vs. with LLM-generated texts from a datastore.

Consequently, we calculate the following metrics for each target span $x_{i:i+n}$: *length score L*, *reliability score R*, and *prediction score P*. The length score $L$ is the number of tokens in the target span:

$$L(x_{i:i+n}) = n \tag{1}$$

The reliability score $R$ is the mean similarity $c_j$ between the target span and each retrieved span:

$$R(x_{i:i+n}) = \frac{\sum_{j=1}^{k} c_j}{k} \tag{2}$$

The reliability score $R$ indicates how strongly the target span is supported by similar spans retrieved from the datastore. The prediction score $P$ is a ratio of LLM label in the original labels $l_j$ of the retrieved spans:

$$P(x_{i:i+n}) = \frac{\sum_{j=1}^{k} \mathbb{1}(l_j = \texttt{LLM})}{k}. \tag{3}$$

The prediction score $P$ indicates whether the target span shares more similar spans with human-written vs. with LLM-generated texts in the datastore.

## 2.2 SPAN SELECTION WITH DP ALGORITHM

In this phase, we select spans $T = [t_1, \ldots, t_H]$ in the target text $x$, so that the text is segmented without overlaps as a final result:

To facilitate users' understanding of the final result, we optimize the span segmentation that includes longer and more similar spans with ones from the datastore. Algorithm 1 describes our dynamic programming strategy to find the best span break. Formally, we select spans $T$ to maximize the score $S$ across the spans in the target text:

$$S(T) = \frac{\sum_{h=1}^{H} \{\alpha L^{\text{std}}(t_h) + (1 - \alpha) R^{\text{std}}(t_h)\}}{H}. \tag{4}$$

Here, $L^{\text{std}}(t_h)$ and $R^{\text{std}}(t_h)$ are the normalized[2] versions of the length score $L$ and the reliability score $R$ of the span $t_h$, respectively. $\alpha$ is an interpolation coefficient ranging from 0.0 to 1.0. $\alpha$ determines the relative contribution of the length score and the reliability score to the span segmentation.

---

[2] To align the scales of the length score and the reliability score, each score is normalized using the mean and the variance in the validation split of our dataset.

---

**Algorithm 1** Span Segmentation Optimization

---

**Input:** Target text $x$; Length of target text $m$; Length score $L$; Reliability score $R$; Maximum length of $n$-gram span $N$; Hyper-parameter $\alpha$
**Output:** List of selected $n$-grams $T$
$\text{dp}[0, \ldots, m-1] \leftarrow [([0], \text{None})] * m$
**for** $i = 1$ **to** $m$ **do**
    **for** $j = \min(i - N, 0)$ **to** $i$ **do**
        $l, r \leftarrow L^{\text{std}}(x_{j:i}), R^{\text{std}}(x_{j:i})$
        $scores \leftarrow \text{dp}[j][0] + [\alpha l + (1 - \alpha)r]$
        $s_{\text{cand}} \leftarrow \text{average}(scores)$
        **if** $\text{average}(\text{dp}[i][0]) < s_{\text{cand}}$ **then**
            $\text{dp}[i] \leftarrow (scores, j)$
        **end if**
    **end for**
**end for**
Traverse $\text{dp}$ backward and collect span breaks
**return** List of selected $n$-grams $T$

---

### 2.3 OVERALL DETECTION WITH EVIDENCE

Given a sequence of the selected spans $T$ each with a prediction score for the target text $x$, ExaGPT identifies a text based on the mean prediction score:

$$P_{\text{overall}} = \frac{\sum_{h=1}^{H} P(t_h)}{H}. \tag{5}$$

ExaGPT classifies a text as `LLM` if $P_{\text{overall}}$ exceeds a detection threshold $\epsilon$, and otherwise as `Human`. As evidence of the decision, ExaGPT provides retrieved top-$k$ similar spans for each span in the text:

$$E = [(t_h, [s_h^1, \ldots, s_h^k])]_{h=1}^{H}. \tag{6}$$

The similarity of the retrieved spans to each span in the target text can help users judge how reliably correct the detection result is.

## 3 EXPERIMENTS

### 3.1 OVERALL SETUP

**Evaluation Measures.** To assess the detection performance, we use the Area Under Receiver Operating Characteristic curve (AUROC) measure, which is widely used in studies on LLM detection. However, it is only useful to observe the overall behavior of a detector through all possible thresholds. In practical scenarios, it is quite important to minimize the false positive classification, i.e., wrongly identifying human-written texts as LLM-generated. We thus report the detection accuracy with a threshold by fixing the false-positive rate (FPR) at 1%, which is an evaluation stream among recent robustness studies (Krishna et al., 2023; Hans et al., 2024; Dugan et al., 2024).

**Datasets.** We use the M4 dataset (Wang et al., 2024), which is a large-scale LLM detection benchmark consisting of pairs of human-written and LLM-generated texts across multiple languages, domains, and generators. In our experiments, we use the English subset, including 3,000 pairs of human-written and LLM-generated texts from each combination of **four domains**: Wikipedia, Reddit, WikiHow, and arXiv, as well as **three generators**: ChatGPT, GPT-4 as closed-source LLMs, and Dolly-v2 (Conover et al., 2023) as open-source LLMs. For each combination, we split the dataset into three parts: train/validation/test with 2,000/500/500 pairs, respectively.

**Baselines.** In our experiments, we compare ExaGPT to **three strong and interpretable detectors** (as detailed in §5): RoBERTa with SHAP (Mitrović et al., 2023), LR-GLTR (Wang et al., 2024), and DNA-GPT (Yang et al., 2023). The first one is a supervised classifier based on RoBERTa[3] (Liu, 2019),

---

[3]https://huggingface.co/FacebookAI/roberta-base

which we fine-tune for LLM detection on our train split. Similarly, we train the LR-GLTR detector on our train split with selected and hand-crafted GLTR features (Gehrmann et al., 2019), following Wang et al. (2024). The hyper-parameter settings for training both RoBERTa and LR-GLTR are aligned with Wang et al. (2024). Configuration details of the baseline detectors are given in Appendix B.

**Settings of ExaGPT.** In the span scoring phase, ExaGPT leverages our train split as the datastore for each combination of domains and generators. We consider the size of $n$-gram to be from 1 to 20 throughout the entire dataset. We embed the target span and all spans in the datastore into the same vector space using BERT-large[4]. For a span embedding, we feed a text into the BERT-large and take the mean second-layer[5] hidden outputs of tokens included in the span. We retrieve the top-$k$ (=10)[6] most similar spans from the datastore for each target span via $k$-NN search using the FAISS library (Johnson et al., 2017). In the span selection phase, we select the optimal $\alpha$ from values between 0.0 and 1.0 at 0.125 intervals, where ExaGPT exhibits the best detection performance in our validation split. The $\alpha$ is constant through our evaluation of the interpretability and the detection performance of ExaGPT.

**Human Evaluation on Interpretability.** We assess the interpretability of the detectors via human evaluation, as it is vital for a good detector to offer interpretable evidence, allowing users to judge how reliably correct the detection result is. Accordingly, we design a human evaluation where participants are provided with detection evidence and judge whether the detection is correct. Therefore, the evaluation metric for interpretability is the accuracy of the human judgments on the detection correctness based on the evidence. For each detector, we evaluate 96 samples[7] from our test split in all combinations of domains and generators so that the ratio of correct and incorrect detections[8] is even. In our human evaluation, four annotators, including one MSc student, one PhD student, and two researchers working in natural language processing, were provided with different samples.

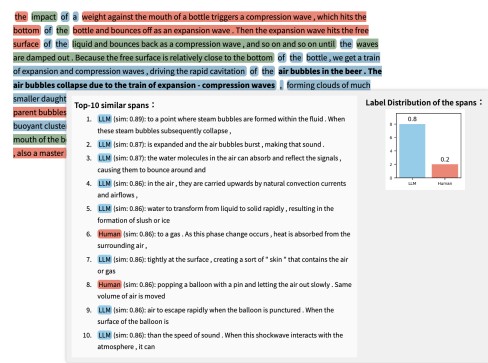

Figure 3: User interface of ExaGPT. Hovering over a text span displays the tooltip about the retrieved similar spans with the similarity to the span and the original label distribution.

Figure 3 shows the user interface of ExaGPT in our human evaluation. The spans are highlighted[9] in red, green, and blue for which prediction score $P$ is lower than 0.5 (human-written), equal to 0.5 (neither), and higher than 0.5 (LLM-generated), respectively. The participants identify the correctness of the detection by mainly investigating similar span examples for each span in the text. We elaborate on the detection evidence of each baseline detector in Appendix C.

## 3.2 MAIN RESULTS

**Detection Interpretability.** Table 1 presents the difference in the accuracy of human judgments on the detection correctness based on evidence across baseline detectors and ExaGPT. The accuracy of human judgments on ExaGPT is relatively higher compared to baseline detectors by up to +13.6 points.

---

[4]https://huggingface.co/google-bert/bert-large-uncased

[5]We select the layer where the $k$-NN spans are similar to the target span well-balanced lexically and semantically, enhancing its interpretability in our pilot study.

[6]We choose the value of $k$ so that ExaGPT shows favorable detection performance over smaller values in our pilot study and does not reduce the interpretability of its evidence. Since ExaGPT presents retrieved spans as evidence, keeping $k$ small helps users assess detection correctness based on a manageable amount of information.

[7]The 96 samples for each detector consist of two samples (one correct and one incorrect) across four domains and three generators, distributed among four participants.

[8]We focus on the setting of the 1% FPR threshold based on practical scenarios.

[9]ExaGPT performs the overall detection rather than detecting each span individually. However, for better readability, each span is color-highlighted on its prediction score.

This indicates that ExaGPT offers more interpretable evidence than other baselines, helping humans judge the correctness of detections more effectively. Here, DNA-GPT also offers $n$-gram span overlaps between the target text and the re-generated LLM texts from the truncated part as evidence. The comparison of the human evaluation score between DNA-GPT and ExaGPT suggests that providing not only simple overlaps but also semantically similar spans contributes to better interpretability. We further investigate how the similarity between the target span and retrieved spans correlates with the correctness of the detection of ExaGPT in §4.

Table 1: **Comparison of interpretability**, as the accuracy (ACC.) of human judgments on the correctness of detections based on evidence across baseline detectors and ExaGPT. Higher accuracy implies that the detector provides more interpretable evidence to users.

| Detector | ACC. of Human Judgements (%) ↑ |
|---|---|
| RoBERTa | 47.9 |
| LR-GLTR | 57.3 |
| DNA-GPT | 53.1 |
| ExaGPT | **61.5** |

**Detection Performance.**  Table 2 shows the difference in the detection performance of baseline detectors and ExaGPT across four domains and three generators. The detection performance includes AUROC and the accuracy at 1% FPR. Overall, ExaGPT consistently demonstrates detection performance on par with or better than baseline detectors, including supervised classifiers. Specifically, on accuracy at 1% FPR, ExaGPT achieves the best average detection performance on all three generators, outperforming baselines by a large margin of up to +37.0 points. This suggests that ExaGPT is the most effective detector in practical scenarios, where we need to minimize the false positives.

**Summary.**  From the experiments on interpretability and classification performance of detectors, we observe that ExaGPT achieves both superior interpretability of the detection and exceptional detection performance compared to previous interpretable detectors.

Table 2: **Comparison of detection performances** of ExaGPT and baseline detectors on texts from various domains and generators. *ACC.* indicates the detection accuracy at 1% FPR. *Avg.* indicates the average performance within each row across domains. **Bold** and Underline indicate the best and runner-up performance for each combination of domains and generators, respectively. ExaGPT achieves the best average detection performance on all three generators in practical scenarios, measured by accuracy at 1% FPR.

| Generator | Detector | Wikipedia | | Reddit | | WikiHow | | arXiv | | Avg. | |
|---|---|---|---|---|---|---|---|---|---|---|---|
| | | AUROC | ACC. | AUROC | ACC. | AUROC | ACC. | AUROC | ACC. | AUROC | ACC. |
| ChatGPT | RoBERTa | **100.0** | 77.1 | **99.8** | 61.0 | **100.0** | 50.0 | **100.0** | 87.3 | **100.0** | 68.9 |
| | LR-GLTR | 95.0 | 60.0 | 99.4 | **94.0** | 97.5 | 85.8 | 99.8 | **97.7** | 97.9 | 84.4 |
| | DNA-GPT | 84.8 | 49.4 | 92.3 | 62.9 | 99.4 | 93.5 | 89.0 | 59.9 | 91.4 | 66.4 |
| | ExaGPT | 98.6 | **92.3** | 98.9 | 86.6 | 99.5 | **96.0** | 99.6 | 95.8 | 99.2 | **92.7** |
| GPT-4 | RoBERTa | **100.0** | 87.8 | **100.0** | 66.4 | **100.0** | 77.4 | **100.0** | 68.6 | **100.0** | 75.1 |
| | LR-GLTR | 97.8 | 85.7 | 99.6 | **97.2** | 94.8 | 77.8 | **100.0** | 98.5 | 98.1 | 89.8 |
| | DNA-GPT | 40.3 | 48.1 | 71.9 | 68.6 | 44.6 | 49.9 | 72.2 | 54.4 | 57.3 | 55.3 |
| | ExaGPT | 98.3 | 87.3 | 99.3 | 91.1 | 98.8 | **92.2** | 99.7 | **98.7** | 99.0 | **92.3** |
| Dolly-v2 | RoBERTa | **100.0** | 61.8 | **100.0** | 50.0 | **100.0** | 70.8 | **100.0** | 82.8 | **100.0** | 66.4 |
| | LR-GLTR | 79.7 | 57.7 | 95.3 | **79.0** | 72.4 | 55.0 | 93.7 | 78.2 | 85.3 | 67.5 |
| | DNA-GPT | 68.0 | 61.5 | 67.5 | 66.1 | 87.7 | **82.3** | 64.9 | 57.7 | 72.0 | 66.9 |
| | ExaGPT | 85.8 | **63.8** | 96.2 | 76.6 | 94.3 | 75.6 | 85.2 | **67.3** | 90.4 | **70.8** |

# 4 ANALYSIS

**What Makes ExaGPT Interpretable.**  Our human evaluations demonstrate that ExaGPT provides highly interpretable evidence for its detection compared to prior detectors. To explore the reason for this, we investigated the difference in the characteristics of the selected spans as a final output between correct and incorrect predictions by ExaGPT. Specifically, we focused on span length and mean similarity between each target span and the retrieved spans (reliability score $R$), which are prioritized in the span selection. We randomly selected 1,000 correct and 1,000 incorrect ExaGPT

Table 3: Examples of $k$-NN spans for a target span retrieved by ExaGPT. The colored part represents the original label for each span (`LLM` in blue and `Human` in red, respectively). In the part of $k$-NN spans, the similarity between the target span and each $k$-NN span is added.

| Target Span | `LLM` | published in 1993. The novel tells the story of a young Jewish slave, Hadassah, |
|---|---|---|
| | `LLM (0.92)` | and was first published in 1936. The book tells the story of three orphaned sisters, |
| | `LLM (0.92)` | published in 2012. The novel revolves around the story of a young woman |
| | `LLM (0.90)` | and published in 2010. The novel tells the story of Michael Beard, a |
| | `LLM (0.90)` | ling of the biblical book, Song of Solomon, and is considered one of the |
| $k$-NN Spans | `LLM (0.90)` | man and published in 1963. The book was later adapted into a Disney film of the |
| | `LLM (0.90)` | . The film tells the story of a young |
| | `Human (0.89)` | the Xanth series. It is the second book of a trilogy beginning with Vale of the |
| | `LLM (0.89)` | published in 1959. The novel is set in the Arctic region and follows the story of Dr. |
| | `Human (0.89)` | . It is the third novel in the Dahak trilogy, after the de |
| | `LLM (0.89)` | for his semi-autobiographical novel, "The Watch that Ends the Night". Born in |

predictions on our test splits across all combinations of domains and generators. Figure 4 presents the reliability score distributions of long spans ($n \geq 10$) in the correct and in the incorrect samples.

A rightward shift indicates that correct samples of ExaGPT include more long spans with higher reliability scores than incorrect ones. From the shift, we empirically observe that offering long spans with high reliability scores helps users judge the correctness of the detections. Table 3 presents examples of long spans ($n = 19$) with high reliability scores for a target span retrieved by ExaGPT. We can see that the retrieved spans are well-balanced, and are lexically and semantically similar to the target span.

**Impact of $\alpha$.** In our experiments, we determined the optimal interpolation coefficient $\alpha$ of ExaGPT (as used in Equation 4), where it exhibits the best detection performance on our validation split. To investigate the robustness of ExaGPT against the choice of $\alpha$, we examine the detection performance variation according to the multiple choices of $\alpha$. Figure 5

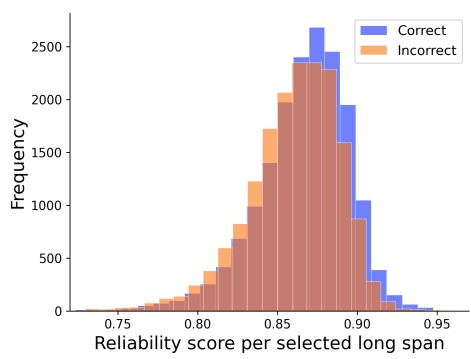

Figure 4: **Reliability score distributions** of long spans ($n \geq 10$) in correct and incorrect samples of ExaGPT, respectively.

depicts the relationship between $\alpha$ and the detection performance of ExaGPT across four domains and three generators: $\alpha$ ranges between 0.0 and 1.0 with 0.125 intervals, and we observe that the higher the $\alpha$, the lower the detection performance. This implies that taking the reliability score more into account (i.e., selecting target spans that are more similar to spans in the datastore) can improve detection performance. On the other hand, across four domains, the lowest performance of AUROC

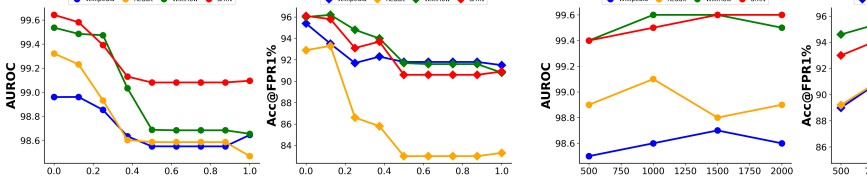

Figure 5: **Impact of $\alpha$** on the detection performance of ExaGPT, including the AUROC and the accuracy at 1% FPR, across four domains using ChatGPT.

Figure 6: **Impact of the datastore size** on the detection performance of ExaGPT, including the AUROC and the accuracy at 1% FPR, across four domains using ChatGPT.

is 98.5%. This suggests that the variation of $\alpha$ in ExaGPT does not lead to its substantial performance drop that could greatly affect the performance ranking of detectors. We find similar overall trends of the impact of $\alpha$ for other LLMs, including GPT-4 and Dolly-v2 as generators. The impact of $\alpha$ on detection performance of ExaGPT in all generators can be found in Appendix D.

**Impact of Datastore Size.** In our evaluation, ExaGPT leverages our train split as the datastore from which it retrieves top-$k$ similar spans for each span in a target text. To explore the robustness of ExaGPT against the size of the datastore, we examine the detection performance variation according to various datastore sizes. Specifically, our train split contains 2,000 pairs of human-written and LLM-generated texts. We randomly sample {500, 1,000, 1,500, 2,000} pairs from our train split as datastores of different sizes.

Figure 6 presents the relationship between the datastore size and the detection performance of ExaGPT across four domains using ChatGPT as a generator. Overall, we find that ExaGPT performs robustly across the size of the datastore with only considerable performance drops. Interestingly, we also observe that ExaGPT with a datastore of 500 pairs, rather than 2,000, achieves comparable detection performance on accuracy at 1% FPR. See Appendix D for consistent trends in all generators, including GPT-4 and Dolly-v2.

## 5 RELATED WORK

**LLM-Generated Text Detection.** Prior studies have presented various types of detection algorithms for LLM-generated texts. They primarily fall into three categories: *text watermarking*, *metrics-based*, and *supervised classifiers*. Text watermarking is a detection approach by calculating the ratio of secret tokens in a target text. Such tokens are randomly selected by a hash function, and their probabilities are intentionally increased at each time step during the LLM decoding process (Kirchenbauer et al., 2023). The metrics-based methods mainly catch the probabilistic discrepancy of a text with the predicted distribution of LLMs. These metrics include token log probabilities (Gehrmann et al., 2019), token ranks (Solaiman et al., 2019; Su et al., 2023b), entropy (Lavergne et al., 2008), perplexity (Beresneva, 2016; Hans et al., 2024), and negative curvature of perturbed text probabilities (Mitchell et al., 2023; Bao et al., 2024). The supervised classifiers are basically models specifically fine-tuned to discern human-written and LLM-generated texts with labeled datasets. The classifiers vary from probabilistic (Ippolito et al., 2020; Crothers et al., 2023) to neural methods (Uchendu et al., 2020; Rodriguez et al., 2022; Guo et al., 2023).

**Interpretability of the Detection Results.** To minimize the undesired consequences of LLM detection (e.g., undermining student's academic dignity), there is need to develop an LLM detector that provides interpretable evidence for the decision. While most detectors output only binary predicted labels, there have been a few studies aiming to provide interpretable evidence. Gehrmann et al. (2019) built a detection tool (called GLTR) that color-highlights tokens in a text with high likelihood under the predicted distribution of LMs. Mitrović et al. (2023); Wang et al. (2024) used explainable machine learning methods, such as LIME (Ribeiro et al., 2016) and SHAP (Lundberg & Lee, 2017), to supervised classifiers. Both explanation approaches basically apply random perturbations to a text and estimate the contribution of each feature to the decision based on the prediction shift. Yang et al. (2023) presented DNA-GPT, a detection method by examining the average ratio of overlapped $n$-gram spans between a truncated target text and multiple LLM-generated continuations. This approach can provide actual LLM-generated texts, including $n$-gram overlaps with the target text as evidence of the detection.

Unlike prior interpretable detectors, our ExaGPT is grounded by the human decision-making process (Maurer et al., 2006; Barrón-Cedeño et al., 2013) of verifying the origin of a text and can provide more interpretable evidence, as explained in the previous sections.

**Example Retrieval for Interpretability.** Beyond the field of LLM text detection, presenting retrieved similar examples has contributed to improving the interpretability of models in various natural language processing tasks. These tasks range from text generation, e.g., machine translation (Khandelwal et al., 2020), to sequential text classification, e.g., part-of-speech tagging (Wiseman & Stratos, 2019), named entity recognition (Jurafsky et al., 2020), and grammatical error correction (Kaneko et al., 2022). At each time step, these methods predict a token or a label from the output

distribution of a base model interpolated with the distribution derived from retrieved nearest neighbor examples.

Our work has a similar direction of using retrieved similar examples for better interpretability with prior studies in other NLP tasks. In LLM text detection, it is particularly crucial to segment the target text into $n$-gram spans for better interpretability, with labels assigned individually (Cheng et al., 2025). Thus, ExaGPT offers a unique mechanism that retrieves similar span examples for each $n$-gram span in the target text and optimizes the final span segmentation based on the examples using dynamic programming.

## 6 CONCLUSION

We introduced ExaGPT, an interpretable human vs. machine detection approach grounded in the human decision-making process of verifying the origin of a text. In particular, ExaGPT classifies a text by examining whether it shares more verbatim and semantically similar spans with human-written vs. with LLM-generated texts from an available datastore. As evidence of the detection, ExaGPT offers similar span examples for each span in the text. The human evaluation and further analysis show that providing similar span examples allows users to judge the correctness of the detection more effectively than prior interpretable detectors. Moreover, extensive experiments in various domains and generators revealed that ExaGPT has shown notably superior detection performance compared to previous strong detectors, even at a false positive rate of 1%. These results indicate that ExaGPT is a detector with both high interpretability in its decision and high detection performance.

## 7 ETHICS AND BROADER IMPACT

**Human Subject Considerations.** In our study, human subjects are engaged in identifying the correctness of the detection based on evidence. All annotators provided informed consent, were fully aware of the study's objectives, and had the right to withdraw at any time.

**Transparency and Reproducibility.** To promote open research, we release our code and data to the public, including all human annotations.

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

## A    LIMITATIONS

**Inference Cost.**    ExaGPT includes a mechanism for retrieving similar spans with each target span from a datastore. In our experiments, the datastore consists of $n$-gram spans ($1 \leq n \leq 20$) from a pair of 2,000 human-written and 2,000 LLM-generated texts. We used four NVIDIA A6000 GPUs to perform the detection within a reasonable time by searching through a vast number of the span instances, which is relatively costly. We could reduce this cost a bit by decreasing the size of the datastore without sacrificing the detection performance (as explained in §4).

**Bias in the Human Judgments.**    Human judgments always carry the risk of subjectivity. Moreover, our evaluation of detector interpretability involves four participants, all of whom are familiar with natural language processing, but in reality, most detector users would not have such expertise. This should be taken into account when interpreting our evaluation results on interpretability.

## B    DETAILED CONFIGURATIONS OF BASELINES

**LR-GLTR.**    Following the setting of Wang et al. (2024), we leverage the two categories of GLTR features: (1) the number of tokens in the top-{10, 100, 1,000, 1,000+} ranks in the predicted probability distribution of LLMs (four features), and (2) the probability distribution of the word divided by the maximum probability of any word at the same position over 10 bins between 0.0 and 1.0 (ten features).

**DNA-GPT.**    For the parameter configuration of DNA-GPT, we set the truncation ratio $\gamma$ to 0.7 and 0.5, and the number of re-generations $K$ to 10 and 5 for closed-source and open-source LLMs, respectively. We also ensured that the `temperature` is the same as the one used to generate a target text and that the generation prompt is known. These configurations were found to ensure the favorable performance of DNA-GPT in Yang et al. (2023). We set all other hyper-parameters to their default values.

## C    DETECTION EVIDENCE OF BASELINES

**RoBERTa with SHAP.**    Figure 7 depicts an example of evidence by RoBERTa with SHAP. We visualize the evidence using the SHAP library[10]. Overall, the red parts are spans that contribute to

---

[10]`https://shap.readthedocs.io/`

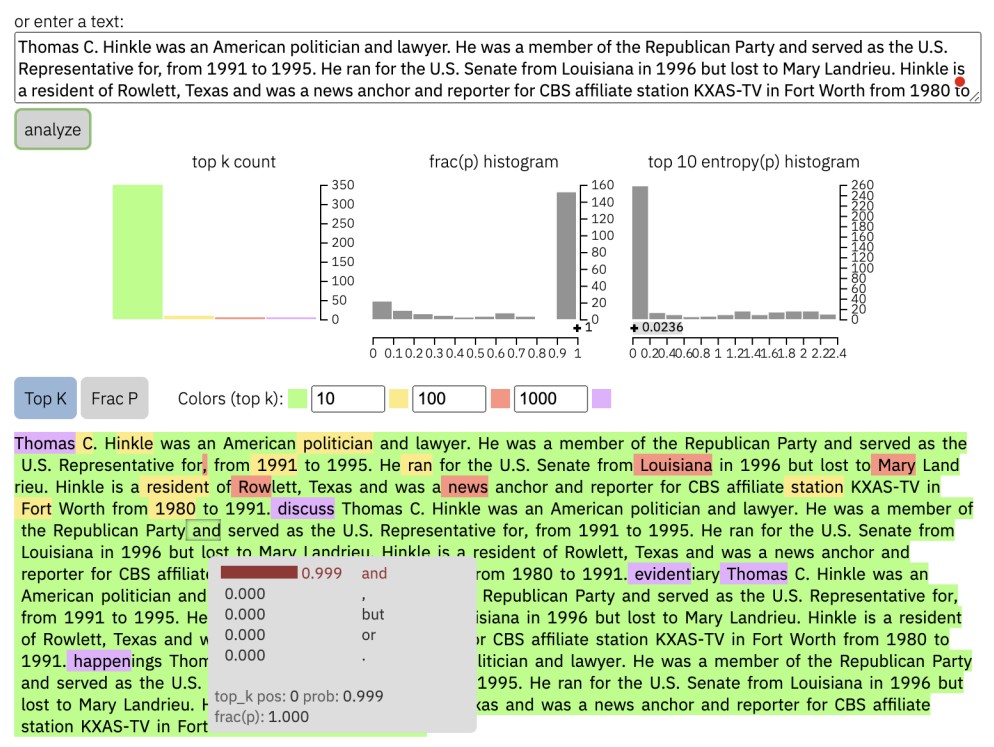

Figure 7: Example of evidence by RoBERTa with SHAP.

predicting LLM-generated. The blue parts are spans that contribute to predicting human-written. In the evidence, if the prediction value, $f(\text{inputs})$ moves further to the right compared to the base value (the expected value across all data samples), it is more likely to be LLM-generated. When we hover over a colored part, we can also see a score of how much the part contributes to the detection result. The more a span contributes to the decision, the darker its color.

**LR-GLTR.** Figure 8 displays an example of evidence by LR-GLTR. We leverage a demo app[11] of GLTR, provided by Gehrmann et al. (2019). It highlights tokens in different colors based on their rank of top-{10, 100, 1,000, 1,000+} in the predicted token distribution from an LLM. The higher the rank of the token, the more likely an LLM is to generate the token. The green parts are spans that an most likely LLM-generated. The degree decreases in the order of green, yellow, red, and purple. When we hover the cursor on a colored part, we can also see the predicted token distribution of an LLM.

Figure 8: Example of evidence by LR-GLTR.

**DNA-GPT.** Figure 9 shows an example of evidence by DNA-GPT. We implemented a demo app of DNA-GPT with the streamlit framework[12]. It shows overlapped $n$-gram spans between a truncated target text and multiple LLM-generated continuations. The more blue spans, the more likely the

---

[11]http://demo.gltr.io/client/index.html
[12]https://github.com/streamlit/streamlit

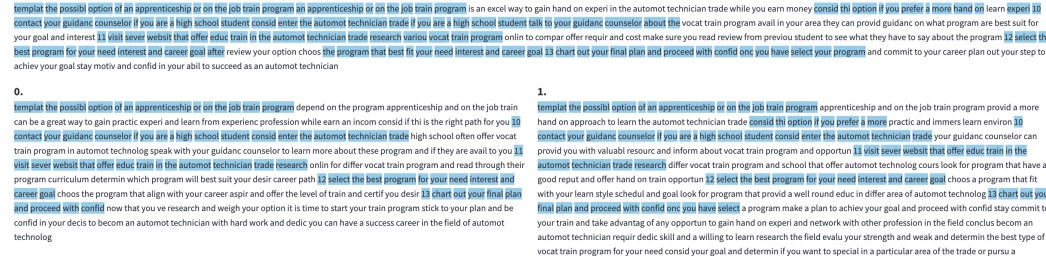

Figure 9: Example of evidence by DNA-GPT.

text is LLM-generated. For span matching, we follow the original implementation of DNA-GPT[13] where it was achieved by token-level matching based on preprocessing of the lower casing and stemming. We also set $n$ to 8 in order to show a large number of overlapped spans enough to interpret as evidence.

# D ANALYSIS DETAILS

**Impact of $\alpha$.** Figure 10 showcases the impact of $\alpha$ on the detection performance of ExaGPT across four domains and three generators. We found similar overall trends of the impact of $\alpha$ in other LLMs, including GPT-4 and Dolly-v2, with the impact in ChatGPT, as explained in §4.

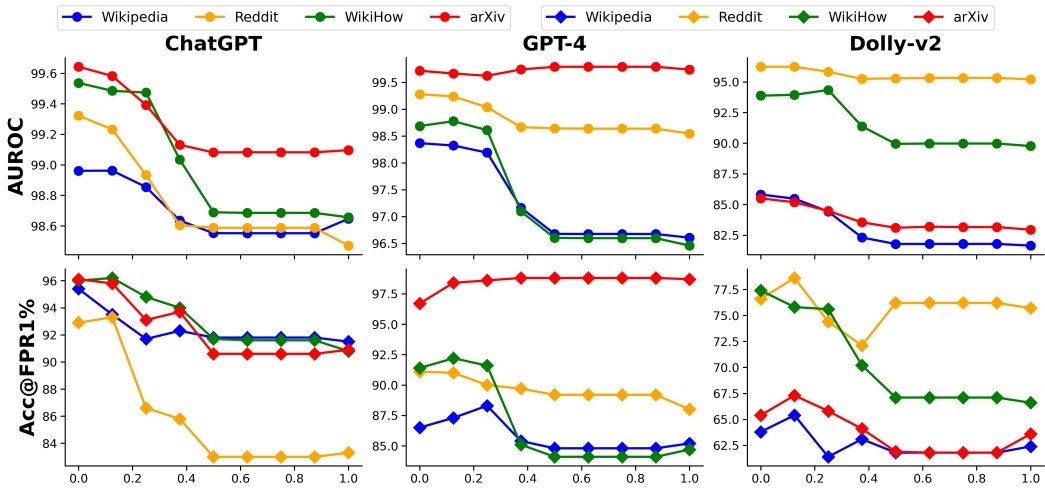

Figure 10: Impact of $\alpha$ on the detection performance of ExaGPT, including the AUROC and the accuracy at 1% FPR, across four domains and three generators.

**Impact of the Datastore Size.** Figure 11 showcases the impact of the datastore size on the detection performance of ExaGPT across four domains and three generators. We can observe similar overall trends of the impact of datastore size in other LLMs, including GPT-4 and Dolly-v2, with the impact in ChatGPT as explained in §4.

---

[13]https://github.com/Xianjun-Yang/DNA-GPT

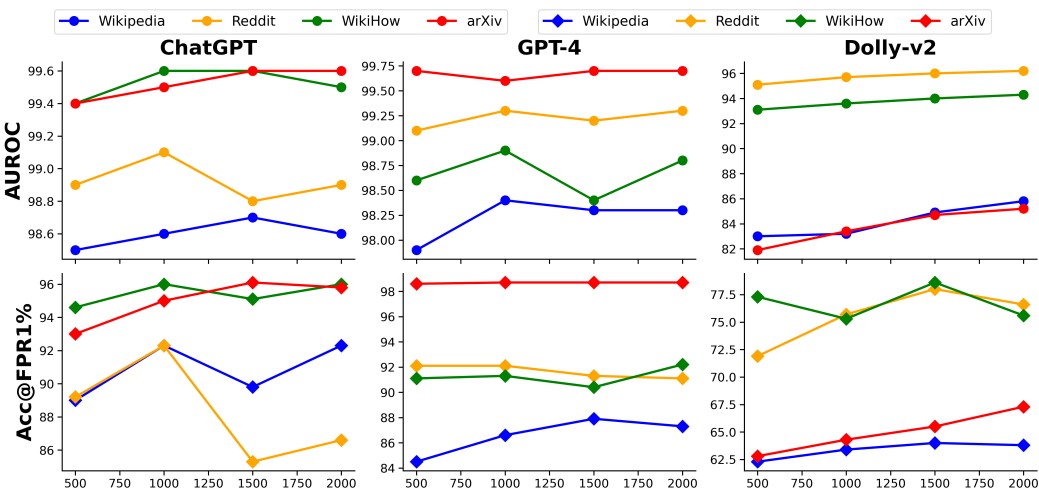

Figure 11: Impact of the datastore size on the detection performance of ExaGPT, including the AUROC and the accuracy at 1% FPR, across four domains and three generators.

