# OpenReview forum: "ExaGPT: Example-Based Machine-Generated Text Detection for Human Interpretability"
_ICLR.cc/2026/Conference — ICLR 2026 Conference Withdrawn Submission_

### Official Review · Reviewer_Eb5c · 2025-10-29

**Soundness:** 2
**Presentation:** 4
**Contribution:** 2
**Rating:** 4
**Confidence:** 3

**Summary:**

This paper  introduces an interpretable detection method that mimics human reasoning when distinguishing between human-written and LLM-generated text. Instead of relying solely on model predictions, ExaGPT compares each span (n-gram) in a target text with similar spans retrieved from a labeled datastore of human and machine examples using k-nearest neighbor search in a shared embedding space (BERT-large). It then uses a dynamic programming algorithm to segment the text into spans that balance length and reliability, providing retrieved examples as transparent evidence for each span.

**Strengths:**

1. The paper is well organized and easy to follow
2. This is a timely and important topic
3. The paper also did a human evaluation study

**Weaknesses:**

1. Datastore dependence is a key vulnerability of ExaGPT. Given that the method relies on retrieving similar spans from a labeled datastore of human-written and LLM-generated texts, both its performance and interpretability are heavily tied to the coverage and quality of this datastore. When the retained instances fail to capture the linguistic variation, writing style, or domain of the target text adequately, the model can pull in irrelevant or misleading spans—leading not only to compromised detection accuracy but also to confusing or uninformative evidence for users. This dependence de facto poses a domain adaptation problem: ExaGPT is ideal when the target text distribution resembles the datastore's, but whether it generalizes to new or niche domains (e.g., scientific abstracts, social media, or non-English data) is uncertain. Moreover, ensuring and curating such a balanced datastore across evolving LLM architectures and writing fashions may grow increasingly time-demanding, possibly limiting ExaGPT's applicability to dynamic real-world environments.
2. The human study is small and biased toward NLP experts, so it’s unclear how interpretable the results would be to non-expert end users. The paper can also include some people without knowledge about NLP for the human study.
3. The method introduces several hyperparameters (span length range, α interpolation, number of neighbors) and I believe this may need dataset-specific tuning or LLM-specific tuning.

**Questions:**

1. How does ExaGPT perform when applied to texts from domains or styles not covered in the datastore? Have you explored domain adaptation or datastore augmentation strategies?

2. The model relies on static BERT-large embeddings for span representation. Would using more recent contextual or instruction-tuned embeddings improve both semantic retrieval and interpretability?

---

> ### Author Response · Authors · 2025-11-23
> **Rebuttal by Authors (1/2)**
>
> Thank you for your valuable feedback. We truly appreciate the time and effort you invested in evaluating our work.
>
> We are glad to see that you value 1) *the organization and clarity of the paper*, 2) *the timeliness and importance of the topic*, and 3) *the inclusion of our human evaluation study*.
>
> We address your concerns below.
>
> ---
>
> ### **Response to Weakness 1: Cross-domain and cross-generator settings**
>
> (This would answer your first question as well.) Thank you for raising this valuable concern. We agree with this and have **investigated how we can effectively leverage ExaGPT in these real-world scenarios**.
>
> We first provide the **cross-domain results** of ExaGPT with GPT-4 as a generator, presented as AUROC / Acc@FPR=1%. The ‘ALL’ indicates the setting where the datastore is constructed from all domains, simply using equal, random samples from each to match the total sample size of the single-domain setting. **Bold** and ***italic*** indicate the best and runner-up performance, respectively.
>
> |  |  |  |  | **Test** |  |  |
> | --- | --- | --- | --- | --- | --- | --- |
> |  |  | **Wikipedia** | **Reddit** | **WikiHow** | **arXiv** | **Average.** |
> |  | **Wikipedia** | **98.3** / **87.3** | 91.7 / 68.2 | 54.1 / 53.3 | 89.3 / 60.5 | 83.4 / ***67.3*** |
> |  | **Reddit** | 90.1 / 60.7 | **99.3** / **91.1** | 74.6 / 50.6 | 93.0 / 63.9 | ***89.3*** / 66.6 |
> | **Train** | **WikiHow** | 66.2 / 50.6 | 76.9 / 60.4 | **98.8** / **92.2** | 64.7 / 51.6 | 76.7 / 63.7 |
> |  | **arXiv** | 73.7 / 50.4 | 86.3 / 56.2 | 57.0 / 51.5 | **99.7** / **98.7** | 79.2 / 64.2 |
> |  | **ALL** | ***94.3*** / ***80.7*** | ***96.7*** / ***83.5*** | ***92.9*** / ***73.4*** | ***99.5*** / ***96.7*** | **95.9** / **83.6** |
>
> Then, we provide the **cross-generator results** of ExaGPT with arXiv as a text domain. ‘ALL’ simply uses a datastore built from all generators, randomly and equally sampled, matching the single-generator total size.
>
> |  |  |  | **Test** |  |  |
> | --- | --- | --- | --- | --- | --- |
> |  |  | **ChatGPT** | **GPT-4** | **Dolly** | **Average.** |
> |  | **ChatGPT** | **99.6 / 95.8** | 98.2 / 84.5 | 63.2 / 50.3 | 87.0 / ***76.9*** |
> | **Train** | **GPT-4** | 94.6 / 66.6 | **99.7 / 98.7** | 61.8 / 51.5 | 85.4 / 72.3 |
> |  | **Dolly** | 93.0 / 69.9 | 89.9 / 65.5 | **85.2 / 67.3** | ***89.4*** / 67.6 |
> |  | **ALL** | ***98.9*** / ***91.4*** | ***99.3*** / ***95.6*** | ***76.4*** / ***52.9*** | **91.5** / **80.0** |
>
> - We observe that when the domain or generator in our datastore is different from the target model or generator, detection performance is reduced. However, in practice, **using multiple sources in the datastore**, instead of relying on only a single source, is considered more realistic. Our results confirm that in such a setting (‘ALL’), ExaGPT consistently maintained **a reasonably high level of performance across multiple domains and generators**.
>
> - Importantly, the tables offer **insights into which domains or generators are effective to consider** for the datastore in terms of generalization. For instance, including **Reddit moderately mitigates the AUROC drop across various domains** or including **ChatGPT helps maintain high detection performance against GPT-4**.
>
> We also note that in real-world scenarios where interpretability is valuable, detection tasks are often motivated by a specific purpose (e.g., identifying AI-written papers in academic settings). In such cases, *the target domain is typically known* and it becomes *critical to construct the datastore to cover dominant models* to ensure reliable performance.
>
> Based on these findings, we look forward to future investigations into optimal datastore configurations for more effective real-world deployment. We will incorporate this exploration and these findings into the final version of the paper.

---

> ### Author Response · Authors · 2025-11-23
> **Rebuttal by Authors (2/2)**
>
> ### **Response to Weakness 2: Selection of annotators**
>
> Thank you for raising this important concern. We acknowledge this limitation, as noted in our paper. We employed four evaluators with NLP backgrounds to **ensure high-quality, consistent feedback** for this initial validation, prioritizing annotation reliability.
>
> However, ExaGPT is structurally **designed to minimize reliance on technical knowledge**. Unlike baselines such as GLTR or SHAP that require interpreting probability distributions or contribution scores, ExaGPT provides **concrete textual evidence**. It **enables users to judge based on intuitive span-level syntactic and semantic similarities**, not domain expertise. Therefore, we believe **the relative advantage of ExaGPT for non-experts would be even greater**.
>
> As for the sample size, while 96 texts is modest, **this evaluation task is inherently information-intensive**. It requires annotators to carefully read and assess the validity of the detection based on its evidence for every single sample. Crucially, our study is **one of the first to test if the evidence *actually helps* users judge whether the detection is correct**. This practical evaluation has been **largely overlooked** in prior works. We believe these findings provide valuable empirical evidence validating the interpretability of our approach as a first step.
>
> We agree that validating this with a broader, non-expert users is an important next step, which we plan to address in future work.
>
> ---
>
> ### **Response to Weakness 3: Need for hyperparameter tuning of ExaGPT**
> Thank you for raising this valuable concern. We believe **extensive tuning is not required**. Empirically, we **used a fixed set of hyperparameters** (span range *n*=1-20, number of neighbors *k*=10) **across diverse domains and generators**, and **consistently achieved high performance**.
>
> - For span range, it is broad enough to **extract interpretable spans ranging from short phrases to full clauses.** This length **roughly corresponds to the average English sentence** (15–20 words), providing **sufficient linguistic coverage without compromising readability**.
> - For the number of neighbors *k*, we choose the value of k so that ExaGPT shows **favorable detection performance over smaller values in our pilot study** and does not reduce the interpretability of its evidence. Since ExaGPT presents retrieved spans as evidence, **keeping k small helps users assess detection correctness** based on a **manageable amount of information**.
> - For the coefficient interpolate *α*, while we selected it based on the dev set performance, **our sensitivity analysis on *α* in Section 4 confirms that performance remains stable** with only marginal variations across a wide range of values.
>
> ---
>
> ### **Response to Question: Possibility of using more recent embedding models**
>
> Yes - we expect that using more recent contextual embeddings would likely improve semantic retrieval performance. However, their impact on interpretability is more nuanced. While they capture abstract semantics better, they might retrieve spans that are semantically similar but lexically distinct, potentially making the evidence less intuitive for users. We plan to investigate this trade-off in future work.

---

> > ### Comment · Reviewer_Eb5c · 2025-11-23
> >
> > Thank you for your rebuttal. It addresses part of my concerns. However, I believe that detailed cross-domain experiments, as well as methods to improve performance in such settings, are crucial for LLM text detection tasks. Therefore, I will keep my original score.

---

### Official Review · Reviewer_Agqe · 2025-10-29

**Soundness:** 2
**Presentation:** 3
**Contribution:** 2
**Rating:** 4
**Confidence:** 4

**Summary:**

This paper proposes ExaGPT, an interpretable AI-generated text detection framework that predicts the score by referencing the k-nearest similar texts in a pre-built datastore. The framework consists of two main stages: span scoring with KNN search and span selection using a dynamic programming algorithm. These stages enable ExaGPT to identify optimal text segmentations and provide interpretable examples for each span, enhancing human understanding of detection results. Evaluations across four text domains and three baseline detectors/generators show comparable detection accuracy of ExaGPT, while human evaluation indicates its higher interpretability.

**Strengths:**

1. ExaGPT enhances interpretability by linking predictions to concrete example spans retrieved from a datastore, offering human-understandable evidence.
2. The proposed method is almost training-free, simplifying deployment in the real-world.
3. The authors provide a visualization interface, improving user accessibility and practical interpretability.

**Weaknesses:**

1. The paper lacks sufficient ablation studies. The influence of factors such as the maximum n-gram size and the choice of text embedding model should be analyzed to understand their effects on detection performance.
2. The method may involve multiple computationally intensive components, including embedding generation, KNN search, and DP-based segmentation, which may significantly increase prediction time. The authors should present an efficiency analysis comparing runtime/throughput against baseline methods.
3. In Figure 3, several retrieved spans appear semantically and lexically different from the target span, reducing interpretability from a human's perspective.
4. The selection of human annotators for the interpretability study raises bias concerns. If the annotators were authors, collaborators, or familiar with the dataset, the evaluation may lack impartiality. A more credible approach would include independent participants or non-expert annotators to better assess usability for the general public.
5. Since the method's core detection capability relies heavily on the datastore's quality, it may perform poorly in cross-domain or cross-model scenarios. The authors should examine ExaGPT's robustness when dealing with paraphrased input, input from an unseen LLM or an unknown text topic/domain.
6. The paper lacks comparison with recent detection baselines, such as [1–3], and should evaluate against more diverse generators as provided in the original M4 dataset.

[1] Guo, Xun, et al. "Detective: Detecting ai-generated text via multi-level contrastive learning." NeurIPS. 2024.

[2] Guo, Hanxi, et al. "Biscope: Ai-generated text detection by checking memorization of preceding tokens." NeurIPS. 2024.

[3] Hans, Abhimanyu, et al. "Spotting LLMs With Binoculars: Zero-Shot Detection of Machine-Generated Text." ICML. 2024.

**Questions:**

Please refer to my questions listed in the Weaknesses section.

**Details Of Ethics Concerns:**

The paper incorporates human annotation results as part of its primary evaluation but does not provide sufficient details on how these studies were conducted. In particular, it is unclear whether the annotation procedures complied with applicable local laws and institutional research-ethics requirements (e.g., IRB or equivalent). Given these uncertainties, I recommend flagging this submission for an ethics review to ensure the human-subjects components meet appropriate ethical and legal standards.

---

> ### Author Response · Authors · 2025-11-23
> **Rebuttal by Authors (1/3)**
>
> Thank you for your valuable feedback. We truly appreciate the time and effort you invested in evaluating our work.
>
> We are glad to see you value 1) *the enhanced interpretability offered by linking predictions to concrete example spans*, 2) *the almost training-free nature of our method that simplifies real-world deployment*, and 3) *the visualization interface designed to improve user accessibility*.
>
> We address your concerns below.
>
> ---
>
> ### **Response to Weakness 1: Ablation studies on maximum *n*-gram size and choice of embedding model**
>
> Thank you for raising your insightful concern. We believe these factors would be **less critical to the core contributions of our work**, due to the following design choices:
>
> - We set the maximum *n*-gram size (*n*=20 tokens) to extract **interpretable spans ranging from short phrases to full clauses**. This length **roughly corresponds to the average English sentence** (15–20 words), providing **sufficient linguistic coverage without compromising readability**.
>
> - We chose BERT as **a widely used text embedding model**. While alternatives like RoBERTa or SBERT might offer marginal gains, we consider exploring them a broader optimization, **orthogonal to our main goal** of **evaluating whether our span-based approach supports interpretability and performance**.
>
> That said, we appreciate the suggestion, and agree that future work could further explore these design dimensions to assess their broader impact on performance and interpretability.
>
> ---
>
> ### **Response to Weakness 2: Inference speed of ExaGPT**
>
> Thank you for raising this important concern. In our preliminary analysis, we found that the inference cost of embedding generation and DP-based segmentation was negligible compared to the ***k*-NN search**, which was the **primary bottleneck due to the large datastore** containing extensive *n*-gram instances.
> Therefore, we have **measured inference latency and GPU memory usage** focusing on the *k*-NN search component. We also have **conducted experiments with FAISS using IVFPQ indexes**, to reduce resource usage and improve inference speed.
>
> Below, we provide the results on the WikiHow domain with ChatGPT. (Here, #Instance refers to the number of n‑gram spans in the datastore.)
>
> |  | #Instance | GPU memory (GB) | Latency (sec) | AUROC |
> | --- | --- | --- | --- | --- |
> | **2000 pair** | 36M | 162.2 | 14.6 | 99.5 |
> | **500 pair** | 9.1M | 54.7 (66%↓) | 5.81 (60%↓) | 99.4 |
> | **500 pair + IVFPQ** | 9.1M | 20.2 (87%↓) | 1.22 (90%↓) | 97.8 |
>
> While we observe that achieving full performance with a datastore of 2,000 pairs requires considerable computational resources and higher latency, **reducing the size to 500 pairs decreases GPU memory usage by 66%** and **latency by 60% without compromising detection performance**. With the additional use of **FAISS‑based *k*-NN approximation**, the requirements are **further reduced by 87% in memory** and **over 90% in latency,** while the **performance drop is still moderate**.
>
> These findings highlight the promising practical applicability of ExaGPT. We will include these additional results and discussion in the Appendix of the final version.
>
> ---
>
> ### **Response to Weakness 3: Different spans observed in Figure 3**
>
> Thank you for the observation. We would like to clarify that Figure 3 is an illustrative example. Due to the **finite coverage of the datastore**, ExaGPT **cannot always retrieve perfectly matching examples** for every single target span.
>
> However, **ExaGPT is designed to provide aggregated evidence across all segmented spans**. Crucially, in our human study, **participants were guided to make judgments based on the overall context** provided by the entire set of retrieved spans, **rather than focusing on local mismatches**. Most importantly, our human study across multiple domains and generators confirms that ExaGPT **offers superior interpretability** to baselines, demonstrating that the **overall evidence remains useful to humans despite occasional mismatches**.

---

> ### Author Response · Authors · 2025-11-23
> **Rebuttal by Authors (2/3)**
>
> ### **Response to Weakness 4: Selection of annotators**
>
> Thank you for raising this important concern. First, we clarify that **the annotators were neither the authors nor collaborators**, and they had **no prior familiarity with the specific dataset used in this study**.
>
> As for their NLP background, we acknowledge this limitation as noted in our paper. We selected the evaluators to **ensure high-quality, consistent feedback** for this initial validation, prioritizing annotation reliability. This choice was made because **the evaluation task is inherently information-intensive**, requiring annotators to carefully read and assess the validity of the detection based on its evidence for every single sample.
>
> However, ExaGPT is structurally **designed to minimize reliance on technical knowledge**. Unlike baselines such as GLTR or SHAP that require interpreting probability distributions or contribution scores, ExaGPT provides **concrete textual evidence** i.e., intuitive span-level similarities. Therefore, we believe **the relative advantage of ExaGPT for non-experts would be even greater**.
>
> Crucially, our study is **one of the first to test if the provided evidence *actually helps* users judge whether the detection is correct**, which is a practical evaluation **largely overlooked** in prior works. We believe these findings provide valuable empirical evidence validating the interpretability of our approach as a first step.
>
> We agree that validating this with a broader, non-expert audience is an important next step for future work.
>
> ---
>
> ### **Response to Weakness 5: Cross-domain and cross-generator settings**
>
> Thank you for raising this valuable concern. We agree with this and have **investigated how we can effectively leverage ExaGPT in these real-world scenarios**.
>
> We first provide the **cross-domain results** of ExaGPT with GPT-4 as a generator, presented as AUROC / Acc@FPR=1%. The ‘ALL’ indicates the setting where the datastore is constructed from all domains, simply using equal, random samples from each to match the total sample size of the single-domain setting. **Bold** and ***italic*** indicate the best and runner-up performance, respectively.
>
> |  |  |  |  | **Test** |  |  |
> | --- | --- | --- | --- | --- | --- | --- |
> |  |  | **Wikipedia** | **Reddit** | **WikiHow** | **arXiv** | **Average.** |
> |  | **Wikipedia** | **98.3** / **87.3** | 91.7 / 68.2 | 54.1 / 53.3 | 89.3 / 60.5 | 83.4 / ***67.3*** |
> |  | **Reddit** | 90.1 / 60.7 | **99.3** / **91.1** | 74.6 / 50.6 | 93.0 / 63.9 | ***89.3*** / 66.6 |
> | **Train** | **WikiHow** | 66.2 / 50.6 | 76.9 / 60.4 | **98.8** / **92.2** | 64.7 / 51.6 | 76.7 / 63.7 |
> |  | **arXiv** | 73.7 / 50.4 | 86.3 / 56.2 | 57.0 / 51.5 | **99.7** / **98.7** | 79.2 / 64.2 |
> |  | **ALL** | ***94.3*** / ***80.7*** | ***96.7*** / ***83.5*** | ***92.9*** / ***73.4*** | ***99.5*** / ***96.7*** | **95.9** / **83.6** |
>
> Then, we provide the **cross-generator results** of ExaGPT with arXiv as a text domain. ‘ALL’ simply uses a datastore built from all generators, randomly and equally sampled, matching the single-generator total size.
>
> |  |  |  | **Test** |  |  |
> | --- | --- | --- | --- | --- | --- |
> |  |  | **ChatGPT** | **GPT-4** | **Dolly** | **Average.** |
> |  | **ChatGPT** | **99.6 / 95.8** | 98.2 / 84.5 | 63.2 / 50.3 | 87.0 / ***76.9*** |
> | **Train** | **GPT-4** | 94.6 / 66.6 | **99.7 / 98.7** | 61.8 / 51.5 | 85.4 / 72.3 |
> |  | **Dolly** | 93.0 / 69.9 | 89.9 / 65.5 | **85.2 / 67.3** | ***89.4*** / 67.6 |
> |  | **ALL** | ***98.9*** / ***91.4*** | ***99.3*** / ***95.6*** | ***76.4*** / ***52.9*** | **91.5** / **80.0** |
>
> - We observe that when the domain or generator in our datastore is different from the target model or generator, detection performance is reduced. However, in practice, **using multiple sources in the datastore**, instead of relying on only a single source, is considered more realistic. Our results confirm that in such a setting (‘ALL’), ExaGPT consistently maintained **a reasonably high level of performance across multiple domains and generators**.
>
> - Importantly, the tables offer **insights into which domains or generators are effective to consider** for the datastore in terms of generalization. For instance, including **Reddit moderately mitigates the AUROC drop across various domains** or including **ChatGPT helps maintain high detection performance against GPT-4**.
>
> We also note that in real-world scenarios where interpretability is valuable, detection tasks are often motivated by a specific purpose (e.g., identifying AI-written papers in academic settings). In such cases, *the target domain is typically known* and it becomes *critical to construct the datastore to cover dominant models* to ensure reliable performance.
>
> Based on these findings, we look forward to future investigations into optimal datastore configurations for more effective real-world deployment. We will incorporate this exploration and these findings into the final version of the paper.

---

> ### Author Response · Authors · 2025-11-23
> **Rebuttal by Authors (3/3)**
>
> ### **Response to Weakness 2: Robustness to paraphrasing attacks**
>
> Thank you for raising this valuable concern. We agree with this and have **investigated the performance of ExaGPT against paraphrasing attack**. We utilized DIPPER [1], which is the 11B document-level paraphraser originally for detection evasion.
>
> Below, we provide the results (AUROC / Acc@FPR=1%) on all domains with ChatGPT.
>
> |  | Wikipedia | Reddit | WikiHow | arXiv | Average. |
> | --- | --- | --- | --- | --- | --- |
> | **LR-GLTR** | 89.4 / 60.2 | 97.0 / **76.8** | 89.5 / 58.0 | **99.6** / **96.7** | 93.9 / 72.9 |
> | **ExaGPT (Ours)** | **98.0** / **86.5** | **97.2** / 69.4 | **91.1** / **73.8** | 97.7 / 76.4 | **96.0** / **76.5** |
>
> Even under paraphrasing attacks, ExaGPT **retains moderately high detection performance**, **outperforming LR-GLTR, which was the runner-up among interpretable detectors** in our original in-domain evaluation. We will include this in our final version.
>
> ---
>
> ### **Response to Weakness 3: Comparison with state-of-the-art non-interpretable detectors**
>
> Thank you for your valuable suggestion. Although **our primary focus is on interpretable detectors**, we agree that comparisons with such methods are **valuable for understanding the trade-off between interpretability and accuracy**.
>
> Given the limited rebuttal period, we focused on methods that can be readily integrated into our evaluation without additional training. We have therefore added **Binoculars** and **Fast-DetectGPT**. Both are **recent and state-of-the-art zero-shot detectors**, and we evaluated them on texts generated by ChatGPT. We employed each method’s default configuration, and report AUROC / Acc@FPR=1%. **Bold** and ***italic*** indicate the best and runner-up performance, respectively.
>
> |  | Wikipedia | Reddit | WikiHow | arXiv | Average. |
> | --- | --- | --- | --- | --- | --- |
> | **Binoculars** | 83.4 / 49.4 | 72.4 / 55.0 | 80.9 / 65.0 | 84.1 / 58.2 | 80.2 / 56.9 |
> | **Fast-DetectGPT** | **99.6** / **98.1** | **99.4** / **94.7** | **_95.8_** / **_85.4_** | **100.0** / **98.1** | **_98.7_** / **94.1** |
> | **ExaGPT (Ours)** | **_98.6_** / **_92.3_** | **_98.9_** / **_86.6_** | **99.5** / **96.0** | **_99.6_** / **_95.8_** | **99.2** / **_92.7_** |
>
> Notably, we observe that ExaGPT, **despite being interpretable**, achieves detection performance **on par with or even better** than **recent state-of-the-art zero-shot detectors**.
>
> In contrast, DeTeCtive and BiScope, as you suggested, are training-based methods that **require additional fine-tuning and a supervised retrieval index**, which is not feasible to add during the rebuttal period. We will clarify these comparisons in the paper.
>
> We will clarify these comparisons in the paper.
>
> ---
>
> ### **Response to Weakness 4: More diverse generators in the original M4 dataset**
>
> Thank you for the valuable suggestion. Our evaluation includes **ChatGPT and GPT-4**, which are far **more widely used in real-world applications** than **other generators in M4 such as davinci-003 or BLOOMz**. While we do not cover all generators in the full M4 benchmark, we believe that including these widely deployed models provides **practically meaningful insights for real-world detection** scenarios. We will clarify this choice in the paper.
>
> ---
>
> Reference:
>
> [1] Krishna et al. Paraphrasing evades detectors of AI-generated text, but retrieval is an effective defense. NeurIPS 2023.

---

### Official Review · Reviewer_fgzx · 2025-10-30

**Soundness:** 3
**Presentation:** 4
**Contribution:** 3
**Rating:** 4
**Confidence:** 5

**Summary:**

This paper introduces ExaGPT, an interpretable LLM-generated text detection method that identifies texts by comparing them against a datastore of human-written and LLM-generated spans. The approach uses k-NN retrieval to find similar spans and applies dynamic programming to optimize span segmentation. The method provides similar span examples as interpretable evidence. Experiments across four domains and three generators show ExaGPT outperforms baseline interpretable detectors by up to +37.0 accuracy points at 1% FPR, while achieving 61.5% accuracy in human evaluation of interpretability (vs. 47.9-57.3% for baselines).

**Strengths:**

The paper presents a novel approach that connects human decision-making processes to technical design. The k-NN span matching with dynamic programming for LLM detection is creative and well-motivated.

The experimental design is solid, with meaningful human evaluation that directly measures interpretability through judgment accuracy. Testing across multiple domains and generators demonstrates robustness. The focus on 1% FPR addresses real-world needs where false positives have serious consequences.

The paper is well-structured with clear explanations and helpful visualizations. The substantial performance gains and superior interpretability make this a valuable contribution to high-stakes detection scenarios.

**Weaknesses:**

1. The human evaluation involves only four annotators who are all NLP experts, so the results may not generalize to typical users. The sample size of 96 texts per detector is relatively small for drawing strong conclusions about interpretability.

2. The paper does not discuss how ExaGPT handles adversarial attacks or paraphrased texts. If an attacker modifies LLM outputs to avoid matching spans in the datastore, the detection may fail.

3. The datastore needs continuous updates as new LLMs emerge. The paper does not address the maintenance cost or how to handle mixed-authorship texts where humans edit LLM outputs.

4. The evaluation dataset is limited to English text across four domains. The paper lacks experiments on multilingual datasets or code generation, which are important use cases for LLM detection. This narrow scope limits understanding of the method's broader applicability.

5. While the paper focuses on interpretable baselines, it would be valuable to compare against state-of-the-art non-interpretable detectors to understand the performance trade-off between interpretability and accuracy. This would help justify whether the interpretability gains are worth potential performance costs, eg, BiScope, etc.

6. The paper does not evaluate cross-domain generalization. All experiments use domain-specific datastores, but real-world scenarios often require detecting texts from unseen domains. Testing how ExaGPT performs when the datastore domain differs from the test domain would strengthen the evaluation.

**Questions:**

See weakness.

---

> ### Author Response · Authors · 2025-11-23
> **Rebuttal by Authors (1/3)**
>
> Thank you for your valuable feedback. We truly appreciate the time and effort you invested in evaluating our work.
>
> We are glad to see that you value 1) *our novel approach connecting human decision-making to technical design via creative *k*-NN span matching and dynamic programming*, 2) *the robust experimental design prioritizing human evaluation and low FPR*, and 3) *the significant contribution to high-stakes detection scenarios*.
>
> We address your concerns below.
>
> ---
> ### **Response to Weakness 1: Selection of annotators**
>
> Thank you for raising this important concern. We acknowledge this limitation, as noted in our paper. We employed four evaluators with NLP backgrounds to **ensure high-quality, consistent feedback** for this initial validation, prioritizing annotation reliability.
>
> However, ExaGPT is structurally **designed to minimize reliance on technical knowledge**. Unlike baselines such as GLTR or SHAP that require interpreting probability distributions or contribution scores, ExaGPT provides **concrete textual evidence**. It **enables users to judge based on intuitive span-level syntactic and semantic similarities**, not domain expertise. Therefore, we believe **the relative advantage of ExaGPT for non-experts would be even greater**.
>
> As for the sample size, while 96 texts is modest, **this evaluation task is inherently information-intensive**. It requires annotators to carefully read and assess the validity of the detection based on its evidence for every single sample. Crucially, our study is **one of the first to test if the evidence *actually helps* users judge whether the detection is correct**. This practical evaluation has been **largely overlooked** in prior works. We believe these findings provide valuable empirical evidence validating the interpretability of our approach as a first step.
>
> We agree that validating this with a broader, non-expert users is an important next step, which we plan to address in future work.
>
> ---
>
> ### **Response to Weakness 2: Robustness to paraphrasing attacks**
>
> Thank you for raising this valuable concern. We agree with this and have **investigated the performance of ExaGPT against paraphrasing attack**. We utilized DIPPER [1], which is the 11B document-level paraphraser originally for detection evasion.
>
> Below, we provide the results (AUROC / Acc@FPR=1%) on all domains with ChatGPT.
>
> |  | Wikipedia | Reddit | WikiHow | arXiv | Average. |
> | --- | --- | --- | --- | --- | --- |
> | **LR-GLTR** | 89.4 / 60.2 | 97.0 / **76.8** | 89.5 / 58.0 | **99.6** / **96.7** | 93.9 / 72.9 |
> | **ExaGPT (Ours)** | **98.0** / **86.5** | **97.2** / 69.4 | **91.1** / **73.8** | 97.7 / 76.4 | **96.0** / **76.5** |
>
> Even under paraphrasing attacks, ExaGPT **retains moderately high detection performance**, **outperforming LR-GLTR, which was the runner-up among interpretable detectors** in our original in-domain evaluation. We will include this in our final version.
>
> ---
>
> ### **Response to Weakness 3: Maintenance cost of datastore**
>
> Thank you for raising this important concern. While we acknowledge the need for continuous updates, we believe the actual maintenance cost would be **relatively low**.
>
> In our case, the dataset used in our experiments includes generation prompts per sample. Therefore, as new LLMs emerge, we can **simply use them to generate texts based on these existing prompts and incorporate** them into the datastore.
>
> Regarding the performance in such **a scenario where the datastore includes multiple LLMs**, **please kindly refer to our response to Reviewer zfmy** (Response to Weakness: Cross-domain and cross-generator settings). It highlights that including multiple LLMs enables ExaGPT to **consistently maintain a reasonably high level of performance across multiple generators**. It also provides **insights into which domains or generators are effective to consider for the datastore** toward optimal datastore configurations.

---

> ### Author Response · Authors · 2025-11-23
> **Rebuttal by Authors (2/3)**
>
> ### **Response to Weakness 4: Cross-domain and cross-generator settings**
>
> Thank you for raising this important concern. We agree with this and have **investigated how we can effectively leverage ExaGPT in such realistic scenario**.
>
> We provide the **cross-domain results** of ExaGPT with GPT-4 as a generator, presented as AUROC / Acc@FPR=1%. The ‘ALL’ setting indicates that the datastore is constructed from all domains, using equal, random samples from each to match the total sample size of the domain-specific setting. **Bold** and ***italic*** indicate the best and runner-up performance, respectively.
>
> |  |  |  |  | **Test** |  |  |
> | --- | --- | --- | --- | --- | --- | --- |
> |  |  | **Wikipedia** | **Reddit** | **WikiHow** | **arXiv** | **Average.** |
> |  | **Wikipedia** | **98.3** / **87.3** | 91.7 / 68.2 | 54.1 / 53.3 | 89.3 / 60.5 | 83.4 / ***67.3*** |
> |  | **Reddit** | 90.1 / 60.7 | **99.3** / **91.1** | 74.6 / 50.6 | 93.0 / 63.9 | ***89.3*** / 66.6 |
> | **Train** | **WikiHow** | 66.2 / 50.6 | 76.9 / 60.4 | **98.8** / **92.2** | 64.7 / 51.6 | 76.7 / 63.7 |
> |  | **arXiv** | 73.7 / 50.4 | 86.3 / 56.2 | 57.0 / 51.5 | **99.7** / **98.7** | 79.2 / 64.2 |
> |  | **ALL** | ***94.3*** / ***80.7*** | ***96.7*** / ***83.5*** | ***92.9*** / ***73.4*** | ***99.5*** / ***96.7*** | **95.9** / **83.6** |
>
> - We observe that when the domain or generator in our datastore is different from the target model or generator, detection performance is reduced. However, in practice, **using multiple sources in the datastore**, instead of relying on only a single source, is considered more realistic. Our results confirm that in such a setting (‘ALL’), ExaGPT consistently maintained **a reasonably high level of performance across multiple domains**.
>
> - Importantly, the tables offer **insights into which domains are effective to consider** for the datastore in terms of generalization. For instance, including **Reddit moderately mitigates the AUROC drop across various domains**.
>
> Lastly, we also note that in real-world scenarios where interpretability is valuable, detection tasks are often motivated by a specific purpose (e.g., identifying AI-written papers in academic settings). In such cases, *the target domain is typically known* and it becomes *critical to construct the datastore to cover dominant models* to ensure reliable performance.
>
> Based on these findings, we look forward to future investigations into optimal datastore configurations for more effective real-world deployment. We will incorporate this exploration and these findings into the final version of the paper.
>
> ---
>
> ### **Response to Weakness 5: Comparison with state-of-the-art non-interpretable detectors**
>
> Thank you for your valuable suggestion. Although **our primary focus is on interpretable detectors**, we agree that comparisons with such methods are **valuable for understanding the trade-off between interpretability and accuracy**.
>
> Given the limited rebuttal period, we focused on methods that can be readily integrated into our evaluation without additional training. We have therefore added **Binoculars** and **Fast-DetectGPT**. Both are **recent and state-of-the-art zero-shot detectors**, and we evaluated them on texts generated by ChatGPT. We employed each method’s default configuration, and report AUROC / Acc@FPR=1%. **Bold** and ***italic*** indicate the best and runner-up performance, respectively.
>
> |  | Wikipedia | Reddit | WikiHow | arXiv | Average. |
> | --- | --- | --- | --- | --- | --- |
> | **Binoculars** | 83.4 / 49.4 | 72.4 / 55.0 | 80.9 / 65.0 | 84.1 / 58.2 | 80.2 / 56.9 |
> | **Fast-DetectGPT** | **99.6** / **98.1** | **99.4** / **94.7** | **_95.8_** / **_85.4_** | **100.0** / **98.1** | **_98.7_** / **94.1** |
> | **ExaGPT (Ours)** | **_98.6_** / **_92.3_** | **_98.9_** / **_86.6_** | **99.5** / **96.0** | **_99.6_** / **_95.8_** | **99.2** / **_92.7_** |
>
> Notably, we observe that ExaGPT, **despite being interpretable**, achieves detection performance **on par with or even better** than **recent state-of-the-art zero-shot detectors**.
>
> In contrast, BiScope, as you suggested, is a training-based method that **require additional fine-tuning**, which is not feasible to add during the rebuttal period.
>
> We will clarify these comparisons in the paper.

---

> ### Author Response · Authors · 2025-11-23
> **Rebuttal by Authors (3/3)**
>
> ### **Response to Weakness 6: Detection on mixed-authorship texts**
>
> Thank you for the valuable point. Since **ExaGPT is a text-fragment-based detection method**, we believe it is **better suited to such mixed-input settings than existing holistic detectors** such as supervised classifiers or metric-based detectors. While empirical evaluation on mixed-authorship texts is **beyond the scope of this current work**, we consider it a promising direction and plan to investigate this thoroughly in future work.
>
> ---
>
> ### **Response to Weakness 7: Detection on multilingual texts and code**
> Thank you for your another valuable point. While we recognize the value of these settings, we believe **our updated evaluation would be sufficiently comprehensive to validate ExaGPT**.
>
> In this rebuttal, we have significantly expanded our experiments to include 1) **cross-domain** and 2) **cross-generator evaluation**, 3) **robustness against paraphrase attack**, 4) **comparisons with state-of-the-art non-interpretable detectors**, and 5) **latency investigations**.
>
> Together with **our original extensive evaluation across four domains and three generators**, as well as **ablation studies on hyperparameters (e.g., the interpolation coefficient α and datastore size)**, we have rigorously evaluated the method's effectiveness. Given this broad empirical validation of our core claims, we leave the investigation into distinct domains (e.g., multilingual text and code) for future work.
>
> ---
> Reference:
>
> [1] Krishna et al. Paraphrasing evades detectors of AI-generated text, but retrieval is an effective defense. NeurIPS 2023.

---

### Official Review · Reviewer_zfmy · 2025-10-31

**Soundness:** 2
**Presentation:** 3
**Contribution:** 2
**Rating:** 4
**Confidence:** 2

**Summary:**

This paper introduces ExaGPT, a method for detecting AI-generated text. The core idea is to determine the source of a given text by retrieving similar text fragments from a database containing both human-written and AI-generated examples. This retrieval process also serves as an interpretable form of evidence for the final judgment. The authors conduct experiments to show that the proposed method performs well in terms of both detection accuracy on specific benchmarks and interpretability.

**Strengths:**

Novel Approach: The application of instance-based retrieval to the problem of AI-generated text detection, with a focus on enhancing interpretability, is a novel and interesting direction.

Emphasis on Interpretability: The paper's focus on improving user trust in detection results is a significant contribution, as this is a critical factor for the practical adoption of AI detection tools.

**Weaknesses:**

The primary weakness of the evaluation lies in its apparent focus on in-distribution settings, which raises concerns about the method's generalization capabilities. As I understand it, the experiments are conducted using text from AI models and domains that are already represented in the retrieval database. This setup does not address the more challenging and realistic scenario where the detector must face a completely new AI model (e.g., a future proprietary or open-source LLM) or content from a novel domain not present in its database. The analogy to plagiarism detection, which must often handle content from unknown sources, seems particularly pertinent here. Without evidence of out-of-distribution robustness, it is difficult to assess the practical scope and reliability of the proposed work in a constantly evolving AI landscape.

**Questions:**

I have a few questions regarding the methodology and evaluation. I would appreciate it if the authors could provide some clarification, and please correct me if my understanding is mistaken.

Regarding the Choice of Text Representation: The paper mentions using the output of BERT's second layer for generating text fragment representations. This strikes me as a somewhat unconventional choice, as later layers are often used for more semantic-rich embeddings. Could the authors provide a brief explanation for this design choice over, for instance, deeper or final layers? A simple ablation study or reference to prior work justifying this choice would be very helpful to clarify the motivation.

Regarding Practical Considerations: The authors acknowledge the computational cost in the appendix. I am curious about the practical challenges of deploying such a large-scale retrieval-based method in a real-world, low-latency setting. Have the authors considered or explored potential optimization strategies (e.g., hierarchical navigation, or other approximate nearest neighbor techniques) to balance the trade-off between detection effectiveness and computational efficiency?

---

> ### Author Response · Authors · 2025-11-23
> **Rebuttal by Authors (1/2)**
>
> Thank you for your valuable feedback. We truly appreciate the time and effort you invested in evaluating our work.
>
> We are glad to see that you value 1) *the novelty of applying instance-based retrieval to AI detection* and 2) *our focus on interpretability as a significant contribution to user trust and practical adoption of AI detectors*.
>
> We address your concerns below.
>
> ---
> ### **Response to Weakness: Cross-domain and cross-generator settings**
>
> Thank you for raising this valuable concern. We agree with this and have **investigated how we can effectively leverage ExaGPT in these real-world scenarios**.
>
> We first provide the **cross-domain results** of ExaGPT with GPT-4 as a generator, presented as AUROC / Acc@FPR=1%. The ‘ALL’ indicates the setting where the datastore is constructed from all domains, simply using equal, random samples from each to match the total sample size of the single-domain setting. **Bold** and ***italic*** indicate the best and runner-up performance, respectively.
>
> |  |  |  |  | **Test** |  |  |
> | --- | --- | --- | --- | --- | --- | --- |
> |  |  | **Wikipedia** | **Reddit** | **WikiHow** | **arXiv** | **Average.** |
> |  | **Wikipedia** | **98.3** / **87.3** | 91.7 / 68.2 | 54.1 / 53.3 | 89.3 / 60.5 | 83.4 / ***67.3*** |
> |  | **Reddit** | 90.1 / 60.7 | **99.3** / **91.1** | 74.6 / 50.6 | 93.0 / 63.9 | ***89.3*** / 66.6 |
> | **Train** | **WikiHow** | 66.2 / 50.6 | 76.9 / 60.4 | **98.8** / **92.2** | 64.7 / 51.6 | 76.7 / 63.7 |
> |  | **arXiv** | 73.7 / 50.4 | 86.3 / 56.2 | 57.0 / 51.5 | **99.7** / **98.7** | 79.2 / 64.2 |
> |  | **ALL** | ***94.3*** / ***80.7*** | ***96.7*** / ***83.5*** | ***92.9*** / ***73.4*** | ***99.5*** / ***96.7*** | **95.9** / **83.6** |
>
> Then, we provide the **cross-generator results** of ExaGPT with arXiv as a text domain. ‘ALL’ simply uses a datastore built from all generators, randomly and equally sampled, matching the single-generator total size.
>
> |  |  |  | **Test** |  |  |
> | --- | --- | --- | --- | --- | --- |
> |  |  | **ChatGPT** | **GPT-4** | **Dolly** | **Average.** |
> |  | **ChatGPT** | **99.6 / 95.8** | 98.2 / 84.5 | 63.2 / 50.3 | 87.0 / ***76.9*** |
> | **Train** | **GPT-4** | 94.6 / 66.6 | **99.7 / 98.7** | 61.8 / 51.5 | 85.4 / 72.3 |
> |  | **Dolly** | 93.0 / 69.9 | 89.9 / 65.5 | **85.2 / 67.3** | ***89.4*** / 67.6 |
> |  | **ALL** | ***98.9*** / ***91.4*** | ***99.3*** / ***95.6*** | ***76.4*** / ***52.9*** | **91.5** / **80.0** |
>
> - We observe that when the domain or generator in our datastore is different from the target model or generator, detection performance is reduced. However, in practice, **using multiple sources in the datastore**, instead of relying on only a single source, is considered more realistic. Our results confirm that in such a setting (‘ALL’), ExaGPT consistently maintained **a reasonably high level of performance across multiple domains and generators**.
>
> - Importantly, the tables offer **insights into which domains or generators are effective to consider** for the datastore in terms of generalization. For instance, including **Reddit moderately mitigates the AUROC drop across various domains** or including **ChatGPT helps maintain high detection performance against GPT-4**.
>
> We also note that in real-world scenarios where interpretability is valuable, detection tasks are often motivated by a specific purpose (e.g., identifying AI-written papers in academic settings). In such cases, *the target domain is typically known* and it becomes *critical to construct the datastore to cover dominant models* to ensure reliable performance.
>
> Based on these findings, we look forward to future investigations into optimal datastore configurations for more effective real-world deployment. We will incorporate this exploration and these findings into the final version of the paper.

---

> ### Author Response · Authors · 2025-11-23
> **Rebuttal by Authors (2/2)**
>
> ### **Response to Question 1: Choice of embedding layer**
>
> Thank you for your valuable question. As briefly noted in the footnote on L225, our decision to use the second layer was **driven primarily by interpretability**. We selected this layer to ensure that the retrieved ***k*-NN spans should be similar to the target span well-balanced lexically and semantically**.
>
> In our pilot study, we observed that using deeper layers resulted in a marginal performance improvement (approximately 1 point in AUROC). However, since BERT gradually abstracts token representations, **deeper layers** tend to be **more sensitive to contexts** or semantic information [1]. Consequently, we observed that the retrieved *k*-NN spans frequently **shared similar surrounding contexts but were lexically very different** from the target span. This mismatch **undermines the intuitive interpretability** of the detection.
>
> Therefore, we decided to use the second layer to **prioritize clear interpretability while ensuring high detection performance**. We would also like to note that the specific embedding choice is not the core contribution of our proposal, thus future work could explore more recent or advanced embedding models to further optimize this component.
>
> ---
>
> ### **Response to Question 2: Optimization strategy for computational efficiency**
>
> Thank you for your another valuable question. We acknowledge that our paper did not explicitly report the required computational resources or inference latency. To further answer this question, we have therefore **measured computational aspects** and **additionally conducted experiments with FAISS using IVFPQ indexes**, as suggested, to **reduce resource usage and improve inference speed**.
>
> Below, we provide the results on the WikiHow domain with ChatGPT. (Here, #Instance refers to the number of n‑gram spans in the datastore.)
>
> |  | #Instance | GPU memory (GB) | Latency (sec) | AUROC |
> | --- | --- | --- | --- | --- |
> | **2000 pair** | 36M | 162.2 | 14.6 | 99.5 |
> | **500 pair** | 9.1M | 54.7 (66%↓) | 5.81 (60%↓) | 99.4 |
> | **500 pair + IVFPQ** | 9.1M | 20.2 (87%↓) | 1.22 (90%↓) | 97.8 |
>
> While we observe that achieving full performance with a datastore of 2,000 pairs requires considerable computational resources and higher latency, **reducing the size to 500 pairs decreases GPU memory usage by 66%** and **latency by 60% without compromising detection performance**. With the additional use of **FAISS‑based *k*-NN approximation**, the requirements are **further reduced by 87% in memory** and **over 90% in latency,** while the **performance drop is still moderate**.
>
> These findings highlight the promising practical applicability of ExaGPT. We will include these additional results and discussion in the Appendix of the final version.
>
> ---
>
> References:
>
> [1] Ethayarajh. How Contextual are Contextualized Word Representations? Comparing the Geometry of BERT, ELMo, and GPT-2 Embeddings. EMNLP-IJCNLP 2019.

---

### Comment · Area_Chair_v4r3 · 2025-11-20
**To AI Review**

Dear authors,

I would like to share an important reminder regarding the review process. Recently, we have noticed that some reviewers may be using AI tools to help generate their reviews. This can lead to low-quality or inaccurate feedback, which is unfair to authors who deserve careful and thoughtful evaluations.

To help maintain fairness, I kindly ask for your assistance: If you believe a review you received was partly or fully generated by AI, and you have some evidence (for example: unusual writing style, clear factual mistakes, AI-detector results, repeated generic sentences, etc.), please feel free to contact me directly.

I will review any evidence you provide and, if appropriate, adjust the weight of the reviewer’s evaluation so that it does not negatively affect your submission. Thank you for helping us keep the review process fair and responsible. Your understanding and cooperation are greatly appreciated.

Best regards,

AC

---

### Comment · Area_Chair_v4r3 · 2025-11-27

Dear Reviewers and Authors,

As we are approaching the rebuttal deadline, I would like to share a gentle reminder with everyone.

For authors:
If you have not yet submitted your rebuttal, please make sure to do so as soon as possible. Submitting very close to the deadline may reduce the chance for reviewers to read and respond in time, which could affect the discussion phase.

For reviewers:
If a rebuttal has already been submitted for your assigned paper, I encourage you to take a moment to read it and, where appropriate, provide a brief response or update your evaluation. Of course, this is not meant to pressure anyone into changing scores, it is simply to ensure that all reviews remain well-informed before final decisions.

Thank you all for your time and effort in keeping the review process smooth and constructive.

Warm regards,
AC

---

### Author Response · Authors · 2025-12-04
**Summary**

We sincerely appreciate all reviewers for their time and feedback. We are encouraged that they recognized the novelty of our k-NN span matching and DP algorithm, the solidity of our experiments including human evaluation, and the valuable contribution our detector’s superior performance and enhanced interpretability make to high-stakes scenarios.

During the rebuttal phase, we carefully responded to all reviewer comments with detailed and comprehensive explanations. Here, we briefly summarize the common concerns and our key clarifications. In addition to the reviewer-specific points, we believe we have sufficiently addressed the major concerns.

**Common concerns:**

- **Cross-(domain | generator) settings.** (zfmy, fgzx, Agqe, Eb5c) While our primary goal is to improve interpretability in AI text detection, we also conducted extensive cross-domain and cross-generator experiments to examine how ExaGPT can be further leveraged in more realistic settings. Our results show that incorporating multiple data sources into the datastore enables ExaGPT to maintain strong performance across diverse domains and generators, while providing practical guidance on which domains and generators to include for a more optimal datastore.
- **Selection of human annotators.** (fgzx, Agqe, Eb5c) We clarified that our study is one of the first in AI text detection to evaluate interpretability in a realistic setting, testing whether the evidence actually helps users judge detection correctness. Since this is the initial validation of such an information-intensive task, we prioritized annotation reliability in selecting the annotators. We also emphasized that the intuitive textual evidence of ExaGPT minimizes reliance on technical expertise, suggesting even greater benefits for non-experts, which we plan to examine in future work.
- **Computational cost and latency.** (zfmy, Agqe) We conducted further experiments showing that cost and latency are acceptably low with a smaller datastore, and can be further reduced with *k*-NN approximation, with only minimal effect on performance.
- **Paraphrasing attack.** (fgzx, Agqe) We additionally conducted experiments showing that ExaGPT retains strong detection performance, with only a moderate drop compared to the baseline, even under paraphrasing attacks.
- **Comparison with state-of-the-art (non-interpretable) detectors.** (fgzx, Agqe) We conducted further experiments showing ExaGPT, despite being interpretable, achieves detection performance on par with or even better than recent state-of-the-art zero-shot detectors.

We deeply appreciate the reviewers’ thoughtful feedback, which has greatly improved our work.

---

### Note · Authors · 2026-01-06

I have read and agree with the venue's withdrawal policy on behalf of myself and my co-authors.